# Automated image analysis to assess hygienic behaviour of honeybees

Gianluigi Paolillo[1], Alessandro Petrini[2], Elena Casiraghi[2,3]*, Maria Grazia De Iorio[1], Stefano Biffani[4], Giulio Pagnacco[4], Giulietta Minozzi[1], Giorgio Valentini[2,3]

1 Dipartimento di Medicina Veterinaria, Università degli Studi di Milano, Lodi, Italy, 2 AnacletoLab—Computer Science Department "Giovanni degli Antoni"—DI, Università degli Studi di Milano, Milan, Italy, 3 CINI National Laboratory in Artificial Intelligence and Intelligent Systems, Rome, Italy, 4 CNR-IBBA Milano, Milan, Italy

* elena.casiraghi@unimi.it

**Data Availability Statement:** All relevant data are available at https://github.com/gpaolillo94/Comb_Images.

**Funding:** Funded study. G.M. received the funding. This work was supported by the BEENOMIX and

## Abstract

Focus of this study is to design an automated image processing pipeline for handling uncontrolled acquisition conditions of images acquired in the field. The pipeline has been tested on the automated identification and count of uncapped brood cells in honeybee (*Apis Mellifera*) comb images to reduce the workload of beekeepers during the study of the hygienic behavior of honeybee colonies. The images used to develop and test the model were acquired by beekeepers on different days and hours in summer 2020 and under uncontrolled conditions. This resulted in images differing for background noise, illumination, color, comb tilts, scaling, and comb sizes. All the available 127 images were manually cropped to approximately include the comb area. To obtain an unbiased evaluation, the cropped images were randomly split into a training image set (50 images), which was used to develop and tune the proposed model, and a test image set (77 images), which was solely used to test the model. To reduce the effects of varied illuminations or exposures, three image enhancement algorithms were tested and compared followed by the Hough Transform, which allowed identifying individual cells to be automatically counted. All the algorithm parameters were automatically chosen on the training set by grid search. When applied to the 77 test images the model obtained a correlation of 0.819 between the automated counts and the experts' counts. To provide an assessment of our model with publicly available images acquired by a different equipment and under different acquisition conditions, we randomly extracted 100 images from a comb image dataset made available by a recent literature work. Though it has been acquired under controlled exposure, the images in this new set have varied illuminations; anyhow, our pipeline obtains a correlation between automatic and manual counts equal to 0.997. In conclusion, our tests on the automatic count of uncapped honey bee comb cells acquired in the field and on images extracted from a publicly available dataset suggest that the hereby generated pipeline successfully handles varied noise artifacts, illumination, and exposure conditions, therefore allowing to generalize our method to different acquisition settings. Results further improve when the acquisition conditions are controlled.

BEENOMIX 2.0 projects funded by the Lombardy Region (FEASR program), PSR 2014–2020 (grant number2016/00361532-G42F16000540002) and PSR (grant number 201801057971— G44I19001910002). The funders had no role in study design, data collection and analysis, decision to publish, or preparation of the manuscript.

**Competing interests:** The authors have declared that no competing interests exist.

## Introduction

Honeybee families exhibit hygienic behavior (HB) when parasitic mites or diseases infest colonies threatening comb broods [1, 2]. Worker bees sense the presence of diseased larvae or pupae resulting in the removal of dead or infected broods from sealed cells. When the amount of worker bees showing hygienic behavior is sufficient, colony-level resistance is achieved [2]. As of today, hygienic behavior is measured by quantifying the rate of removal of dead broods of a colony. In literature, two main tests are reported: the pin killed brood assay, which implies the physical killing of brood through a needle, and the freeze killed brood (FKB) assay which implies killing of brood by liquid nitrogen [3–5]. Both methods require limited loss of brood area in the hive to measure the amount of dead brood removal from worker bees in a time interval (i.e. 24h). Beekeepers, addressing brood production, either evaluate or manually count brood quantity in the hive. This method is labor-intensive, time-consuming and prone to error. In this regard, semi-automatic or automatic tools could provide a better way for assessing colony health, making use of the progress made in digital photography. In comparison to manual inspection of comb cells, automated evaluation of comb images yields more solid data and grants reproducibility. A variety of semi-automatic tools for evaluating colony health by means of digital images of comb frames have been developed over the years [6–13]. Recent works measured brood area in comb frames through semi-automatic methods such as Photoshop [6] or ImageJ [7], which allowed segmentation in a semi-supervised approach. Subsequent research allowed counting the number of capped brood cells, rather than the quantification of the overall capped brood area, by using ImageJ [8] or the Circle Hough Transform to detect the cells [9]. Recently, a method able to detect and count capped brood cells through circular convolution has been proposed and validated [10]. Many software packages able to evaluate comb frames are available. Some of them [11, 12] perform statistical analysis to study the condition of honeybee colony by using a commercial software ("IndiCounter, WSC Regexperts" available at https://wsc-regexperts.com/en/software-and-databases/ software/honeybee-brood-colonyassessment-software/) which seems to be designed for large scale studies where specific acquisition conditions are often used. On the other hand, the semi-automated pipeline introduced by Jeker et al. [13] seems to requires a laborious acquisition setting that depends on several camera parameters to be carefully set. HoneyBeeComplete displays the classification of capped brood cells with a detection rate of 97.4% [14]; its promising results motivate their usage during subsequent studies [15]; HiveAnalyzer shows the ability to classify other cells in addition to capped brood through linear Support Vector Machines (SVM) with a classification rate of 94% [16]; CombCount displays the detection of both capped brood and capped honey although a user is required to discriminate between the two with selection tools [17]. Recently, a completely automatic tool using convolutional neural networks (CNNs), DeepBee, showed the classification of seven different comb cell classes with a detection rate of 98.7% [18].

Though promising, all the methods are developed on images acquired under controlled acquisitions. This results in ad-hoc techniques developed for handling specific illumination and exposure conditions, therefore hampering generalizability and applicability to different settings. We developed an automatic tool able to automatically count capped brood cells from images acquired by beekeepers after the FKB test to aid in the assessment of the hygienic behavior of the colonies under study. This work derives from the knowledge of recent studies applying digital photography to detection of capped brood in comb frames in the hive. In this work we propose a semi-automated image processing system that is robust against several issues caused by uncontrolled illumination conditions. The model has been developed by exhaustively testing several alternative image processing algorithms, for which a grid search

procedure has been employed to both define the best setting, and to test their robustness with respect to the modification of the optimal values.

The paper is organized as follows: in Section 1 materials and methods are presented; Section 2 reports the experimental results, which are discussed in Section 3.

## Materials and methods

In this study, a digital camera Sony DSC-W810 was used, with the following settings: aperture —3,62; ISO—100; shutter speed 1/50; auto-focus—on, flash -no; compression—JPEG. The images had a resolution of 20.1MPixels (5152x3864px). Using these settings, after using liquid nitrogen and following a time interval (24h) during the FKB, 127 comb area images were photographed. The 127 images were then manually cropped to include the comb area and were used to compose a training set, $I_{Train}$, by randomly extracting 50 images, and a test set, $I_{Test}$, containing the remaining 77 images. After manual cropping, all the images in $I_{Train}$ and $I_{Test}$ have a horizontal x vertical pixel size approximately equal to 5000 x 4000.

Further, to validate our model against a publicly available dataset obtained with a different equipment and under different acquisition conditions, 100 images were randomly selected among those used in [18] (available at https://github.com/AvsThiago/DeepBee-source/archive/release-0.1.zip), and we compared the results obtained when processing them to those obtained on the images in $I_{Test}$, which were more cautiously cropped by experts to strictly include the comb area.

The developed system includes a preprocessing step, described in Subsection Preprocessing, that removes noise and applies a color enhancement and normalization while simultaneously recovering from bad illumination conditions, and a cell segmentation step, described in Cell segmentation and counting, that automatically identifies and counts the cells.

The system has been implemented by using the Python programming language (version 3.7) and the image processing algorithms are imported from the python OpenCv3 v.4.0 package (last upgraded on the 1st of August 2021).

### Preprocessing

In this Section, we describe the image pre-processing steps we consecutively applied to reduce the effects of gaussian and salt and pepper noise due to the image acquisition equipment, and to harmonize the illumination conditions and background colors in the images, whose variability is due to the uncontrolled acquisitions in different days and times of the day. More precisely, while the salt and pepper and Gaussian noise reduction problem was addressed by the application of a classic digital image processing procedure, where a median filter (3x3 support size) is followed by a 3 x 3 Gaussian filter (standard deviation $\sigma = 5$), to recover from not-uniform or poor illumination conditions and/or varied background colors we comparatively evaluated three different image enhancement algorithms [19–23]. Among them the *Automatic Color Equalization* algorithm, alias ACE [19–21], is based on the principles of human perception and has been successfully used in several fields, among which image and movie restoration, where it has been used for both color and poor illumination restoration, and underwater imaging, where it was used for image dehazing. The image enhancement results produced by ACE have been compared to the image harmonization produced by two algorithms, Macenko's [22] and Vadahane's [23] algorithms, generally exploited in the field of digital immuno-histology for harmonizing the differing bio-marker staining colors due by acquisitions in different days and by different human operators. In the following, we provide a detailed description of the aforementioned algorithms.

For simultaneously handling poor illuminations and differing color conditions, several color normalization algorithms have been experimented, ranging from unsupervised color-enhancement models to color normalization techniques used in digital histology. Regarding unsupervised color-enhancement models, spatial color algorithm (SCA) called Automatic Color Equalization (ACE) was tested, which can adjust the contrast of the image to approximate the color constancy and brightness constancy of the human eye [19–21] (The Automatic Color Equalization method used is implemented in colorcorrect v.0.9.1 python module). Furthermore, two color normalization techniques mostly used in digital histology were applied, which are an algorithm developed by Macenko et al. [22] and the structure-preserving color normalization algorithm (SPCN) presented by Vahadane et al. [23], which allow normalizing the color of histopathological images stained with Hematoxylin-Eosin under different acquisition conditions (The implementation of both Macenko's [22] and Vadahane's [23] methods is available in the python staintools v.2.1.2 package). The last two color-normalization methods described, modify the color characteristics in a set of images so as to make it as similar as possible to the color characteristics of a target image used as reference.

After the normalization step, images were re-scaled to a 10% of their original size to cut processing times for the detection step, and were then converted to grayscale images (RGB to grayscale conversion was performed by using the cvtColor function of OpenCV).

## Cell segmentation and counting

Uncapped cells in comb images appear as dark spots or holes surrounded by a lighter, quasi-circular border; this characteristic is highlighted by the image enhancement step applied in the pre-processing phase (see Fig 1 - image normalization box, and Fig 2). Given this peculiarity, the automatic count of uncapped cells, may be performed by applying a first step that automatically separates the dark areas from the lighter borders; next, all the identification of the individual cells may be performed by processing the areas corresponding to light borders, to identify (and then count) those areas that correspond to circles with a proper size. The first step may be solved by applying an image binarization algorithm; to this aim, we tested different methods, all described in detail in Section Image binarization methods. On the other hand, the second step can be performed by scanning the image to search for shape patterns that may be approximated by circles. This may be done by exploiting the Hough Transform (see Section Circle detection by Hough Transform), a classic image processing Transform used to detect circles in images.

## Image binarization methods

At this stage, the gray level image is binarized by using three different approaches, the first of which is the Otsu's automatic thresholding method [24], a parameter free algorithm that finds the optimal gray level threshold that enables to classifying the image pixels in two classes, by minimizing the intra-class gray level variance, while simultaneously maximizing inter-class gray level variance.

To have a benchmark for comparison, the results obtained by Otsu's algorithm were compared to the Adaptive Mean Thresholding (AMT) method, which selects a pixel if the difference between its gray level and the mean gray level of its neighborhood (with radius blocksize) is greater than a constant C, and the Adaptive Gaussian Thresholding (AGT) method, which works as the AMT but substitutes the row mean of the neighborhood pixels with a weighted mean, where the weights are those of a gaussian centered at the pixel itself and standard deviation equal to 0.3*((blocksize-1)*0.5–1)+0.8 (The AMT, AGT, and Otsu's methods are available

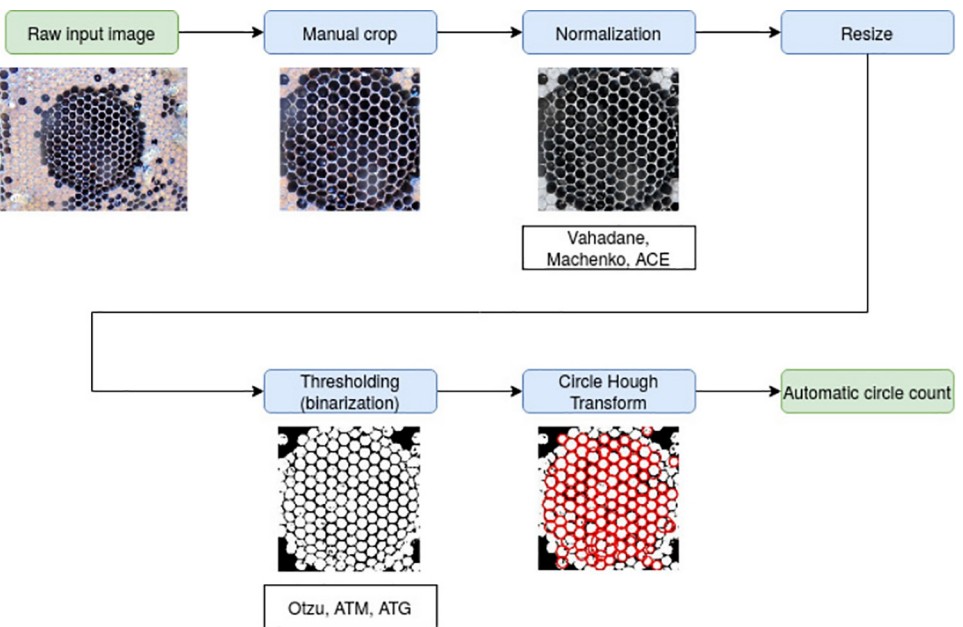

**Fig 1. Image analysis pipeline steps overview.** A raw input image (5152x3864) is manually cropped to extract the circular region of interest (2243x2250) of the FKB. Sampled image is normalized and, then, resized to a 10% of its original dimension (224x225); the resized image is used to generate a binary image which is, then, fed to the Circle Hough Transform for uncapped honeycomb cell detection.

from the opencv-python 4.5.1.48 library). Since AMT and AGT require two critical parameters to be defined, we fine-tuned them through a grid search approach.

## Circle detection by Hough Transform

The Hough Transform was developed to detect lines in images [25]; in practice, given a set of lines to be detected in a binary, and considering that each line y = f(x) can be alternatively expressed through polar coordinates as y = f(R, Θ) = R cos Θ (where R is the radius and theta is the line orientation), it constructs an accumulator matrix where one of the two dimensions are indexed, respectively, by the possible values of theta and of the radius. Next, for each pixel p(x,y) set to 1 in the input image it increases the elements (r, theta) in the accumulator matrix if the point is lying on the line, that is x = rcosθ and y = rsinθ. After scanning all the pixels in the image the highest values in the accumulator matrix correspond to all the lines in the image. Considering that a circle centered at point (a,b) in the Euclidean coordinate system is expressed as $(x − a)^2 + (y − b)^2 = R^2$, by using a 3D accumulator matrix that stores all the possible values for the x-coordinate of the center, the y-coordinate for the center, and the radius, the Hough Transform can be easily extended to the detection of circles. In practice, the Circle Hough Transform (CHT) method uses a voting procedure to measure the probability that a region of pixels forms a circle. The implementation used is found in OpenCv3 v.4.0 library and depends on several parameters; we used the default ones for all but for the minimum circle radius, and maximum circle radius, for which we used a grid search, detailed in Section Results, to detect the optimal values.

## Results

In this section we detail all the experiments we performed to select the best performing algorithms for all the steps described in the "Methods" section and their optimal parameter values.

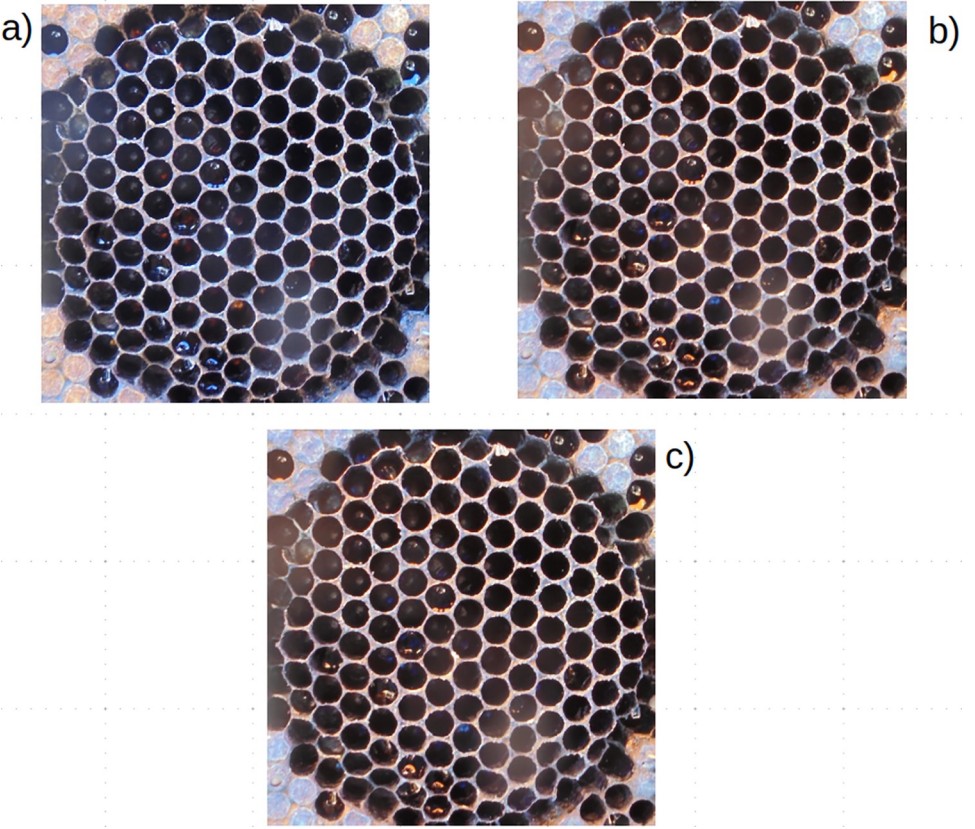

**Fig 2. Noise removal.** (A) Original image, (B) Median Filter, (C) 3x3 Gaussian Filter.

More precisely, we describe the choice of the image enhancement algorithm to be used in the pre-processing phase, and the choice of the image binarization algorithm, which are the preliminary steps before the application of the Hough Transform for Circle (individual uncapped cell) detection. Note that, to perform an unbiased evaluation, the choice between all the candidate algorithms as well as their optimal parameter setting are performed on the training set, composed of 50 images randomly extracted from our dataset. The evaluation of the final pipeline and its best parameter setting is then carried out by using the test set, which consists of images never used during the algorithm selection and the parameter tuning phase. In the following we describe all the technical details of the experiments we performed to choose the best algorithms and parameters, and we motivate our choices by reporting the detailed results we obtained. All the results are summarized and discussed in section "Discussion".

## Cell detection pipeline optimization

Performance of the developed pipeline was assessed on a set of 127 images sampled from a single apiary investigating hygienic behavior. These images showed two hive frame conditions (capped, uncapped brood) and differing lighting conditions, acquisition angles, texture, color conditions, resolution. Images were manually cropped to include the comb area to extract the circular region of interest (2243x2250) of the FKB and were used to compose a training set, $I_{Train}$, by randomly extracting 50 images and a test set, $I_{Test}$, containing the remaining 100 images. After selection of images, the sampled image is, first, denoised, then, normalized and, ultimately, resized to a 10% of its original dimension (224x225). The resized image is used

to generate a binary image which is, then, fed to the CHT for uncapped honeycomb cell detection. An overview of the generated pipeline is shown in Fig 1. To estimate pipeline performance, the correlation of automatic measurements of uncapped comb cells with manual counted uncapped cells was used. Each step of the pipeline was tested, progressively, to opt for the best performing algorithms.

### Background noise reduction

At first, to denoise selected images, a median filter was used to remove salt and pepper noise, followed by a 3x3 Gaussian filter to remove Gaussian noise. An example of denoised images is shown in Fig 2.

### Normalization

The first step of the pipeline, from the original image (Fig 3A), produced a normalized image which handled poor illumination and differing lighting conditions; here, three normalization approaches were tested: two color normalization algorithms used in digital histology developed by Vahadane (Fig 3B) and Macenko (Fig 3C) respectively and a spatial color processing method called Automatic Color Equalization (ACE, Fig 3D). To define the best normalization approach, the pipeline was run for each of the three different normalization methods followed by a 10% image resizing, Otsu's automatic thresholding method, which does not require additional parameters, and the CHT with this set of parameters are reported in Table 1. The first set of parameters was: internal accumulator size– 1, minimum distance– 20, Canny threshold– 50, minimum number of votes– 30, minimum radius– 1, and maximum radius – 25. The same pipeline was also run with the resizing step preceding the normalization step as well as a case in which the normalization step was not applied. The obtained correlations are reported in Table 2.

### Thresholding and Circle Hough Transform

The second part of the pipeline is the thresholding step. The output of this step is a binary mask. As the quality of the solution of the subsequent steps strongly depend on this stage, three thresholding approaches were tested and compared: Otsu's automatic thresholding algorithm, which does not require any parameter, and two Adaptive thresholding algorithms, the Adaptive Mean Thresholding (AMT) and the Adaptive Gaussian Thresholding (AGT), whose results depend on two parameters (*blocksize* and C). In particular, both the Adaptive Thresholding algorithms select pixels whose value is greater than the mean or the gaussian-weighted sum of the neighbourhood with size *blocksize* minus a constant C.

After the thresholding phase, Circle Hough Transform (CHT) detects and counts uncapped cells that have a radius in range (*minRadius*, *maxRadius*).

The pipeline composed by OTSU thresholding followed by Circle Hough Transform requires the optimization of two parameters *minRadius* and *maxRadius*, which was performed by grid search in the range (1–75) for both parameters.

For what regards the pipeline composed by any Adaptive Thresholding algorithms followed by Circle Hough Transform, to avoid impractical computational costs, we applied a hierarchical grid search optimization to search the optimal parameters values for (*blocksize*, C, *minRadius*, *maxRadius*). In particular, the whole search space is initially coarsely spanned to find an approximate 'optimal space', where a subsequent grid search is applied to find the optimal parameter values.

Note that, all the pipelines are applied to the images normalized by ACE since this method yielded the highest performance in the normalization step (see section *Normalization*).

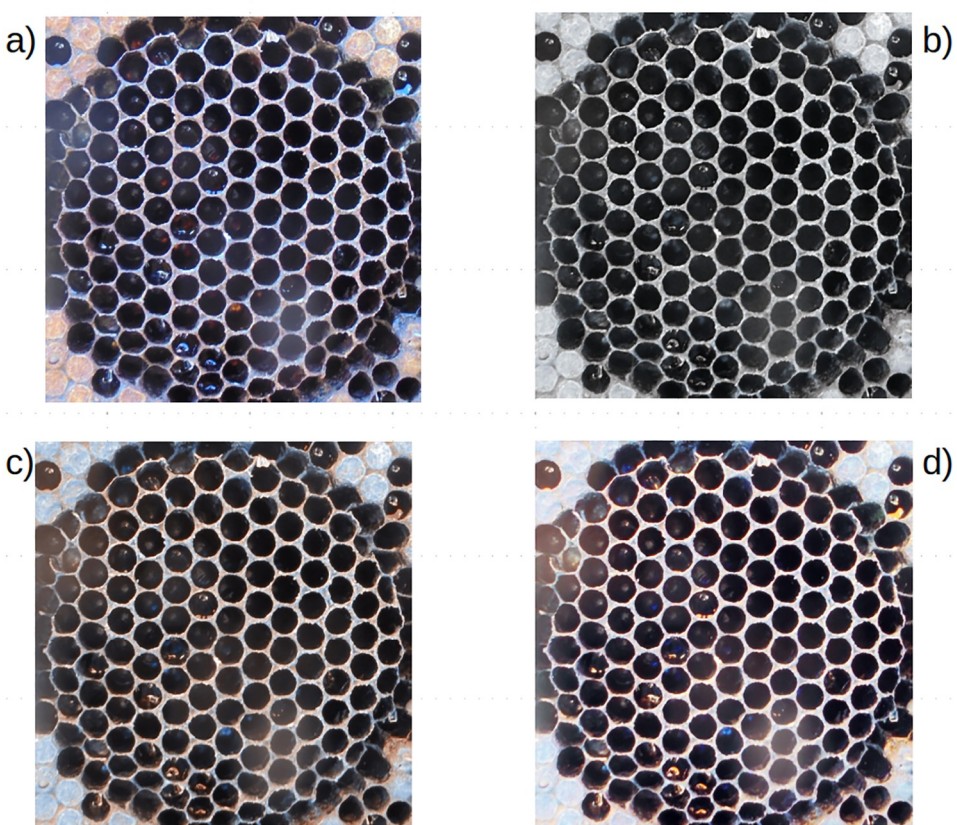

**Fig 3. Normalization test.** (A) Original image, (B) Vahadane method, (C) Macenko method, (D) Automatic Color Equalization.

For OTSU thresholding, a grid search approach was performed on $I_{Train}$ with *minRadius* and *maxRadius* ranging from 1 to 75 with a step of 1 (Fig 4). The highest correlation resulted from: *minRadius*—7, *maxRadius*—29 with a correlation of 0.856. Correlation values distribution for best parameters of *minRadius* and *maxRadius* are shown in Fig 4. Then, the pipeline was tested on $I_{Test}$ with *minRadius* set to 7 and *maxRadius* set to 29 yielding a correlation of 0.819.

For the adaptive thresholding methods, a first grid search approach was performed on $I_{Train}$ with *minRadius* set to 7 (best value found in OTSU), *maxRadius* ranging from 15 to 45 (range was chosen since *maxRadius* was shown, in OTSU in Fig 4, to follow a Gaussian distribution with a plateau around value 29, best value found in OTSU) with a step of 1, *blocksize*

**Table 1. Set of parameters used in the Circle Hough Transform.**

| Parameter name | Value |
| --- | --- |
| Internal accumulator size | 1 |
| Minimum distance | 10 |
| Canny threshold | 25 |
| Minimum number of votes | 15 |
| Minimum radius | 1 |
| Maximum radius | 25 |

Table 2. Pipeline performance in normalization step.

| Step 1 | Step 2 | Correlation % |
| --- | --- | --- |
| Vahadane | 10% Resizing (214x225) | 0.797 |
| Macenko | 10% Resizing (214x225) | 0.757 |
| ACE | 10% Resizing (214x225) | 0.825 |
| 10% Resizing (214x225) | Vahadane | 0.481 |
| 10% Resizing (214x225) | Macenko | 0.517 |
| 10% Resizing (214x225) | ACE | 0.439 |
| No Normalization | 10% Resizing (214x225) | 0.748 |

ranging from 3 to 50 with a step of 4, $C$ ranging from 1 to 60 with a step of 4. The highest correlation resulted from: for the AMT method, *maxRadius*—24, *C* constant—36, *blocksize*—7 with a correlation of 0.855 on $I_{\{Train\}}$ and a correlation of 0.801 on $I_{\{Test\}}$; for the AGT method, *maxRadius*—23, *C* constant—32, *blocksize*—15 with a correlation of 0.864 on $I_{\{Train\}}$ and a correlation of 0.779 on $I_{\{Test\}}$.

A second grid search approach was performed with AMT on $I_{\{Train\}}$ with *blocksize* set to 7 and *C* set to 36 (best values found in previous step) and with AGT on $I_{\{Train\}}$ with *blocksize* set to 15 and *C* set to 32, while *minRadius* and *maxRadius* both ranged from 1 to 50 with a step of 1., while *minRadius* and *maxRadius* both ranged from 1 to 50 with a step of 1. The highest correlation resulted from: for the AMT method, *minRadius*—5, *maxRadius*—22, *C* constant—36, *blocksize*—7 with a correlation of 0.861 on $I_{\{Train\}}$ and a correlation of 0.803 on $I_{\{Test\}}$; for the AGT method, *minRadius*—4, *maxRadius*—21, *C* constant—32, *blocksize*—15 with a correlation of 0.881 on $I_{\{Train\}}$ and a correlation of 0.780 on $I_{\{Test\}}$.

A global grid search, involving all four parameters, was performed limiting parameters range to a specific subspace. In particular, in AMT, *minRadius* was set to range from 1 to 9 with a step of 1, *maxRadius* was set to range from 17 to 25 with a step of 1, *blocksize* was set to range from 5 to 13 with a step of 2, *C constant* was set to range from 24 to 42 with a step of 2; in AGT, *minRadius* was set to range from -1 to 7 with a step of 1, *maxRadius* was set to range from 16 to 24 with a step of 1, *blocksize* was set to range from 13 to 21 with a step of 2, *C constant* was set to range from 18 to 36 with a step of 2. The highest correlation resulted from: for the AMT method, *minRadius*—5, *maxRadius*—21, *C* constant—33, *blocksize*—9 with a correlation of 0.902 on $I_{\{Train\}}$ and a correlation of 0.834 on $I_{\{Test\}}$; for the AGT method, *minRadius*—3, *maxRadius*—20, *C* constant—27, *blocksize*—17 with a correlation of 0.893 on $I_{\{Train\}}$ and a correlation of 0.787 on $I_{\{Test\}}$. Setting both *minRadius* and *maxRadius* allowed us to show distribution values when comparing C and *blocksize* as shown in the respective surface plots.

*minRadius* correlation values distribution when fixing *maxRadius* to the best value obtained (AMT—21, AGT—20) is reported in Fig 5 as well as *maxRadius* correlation values distribution when setting *minRadius* to the best value obtained (AMT—5, AGT—3) using in both cases fixed C constant and *blocksize* best values (AMT—33–9, AGT—27–17).

The obtained best correlations are reported in Table 3.

## Comparative analysis

To test our pipeline on publicly available images acquired by a different equipment and acquisition strategy, we randomly extracted 100 images from the data-set of comb frames recently published by [18]. To obtain the manual counts, a rectangular area was manually cropped (Fig 6A). The cropped images were then used to compose a training set, $I_{\{Train\}}$, by randomly

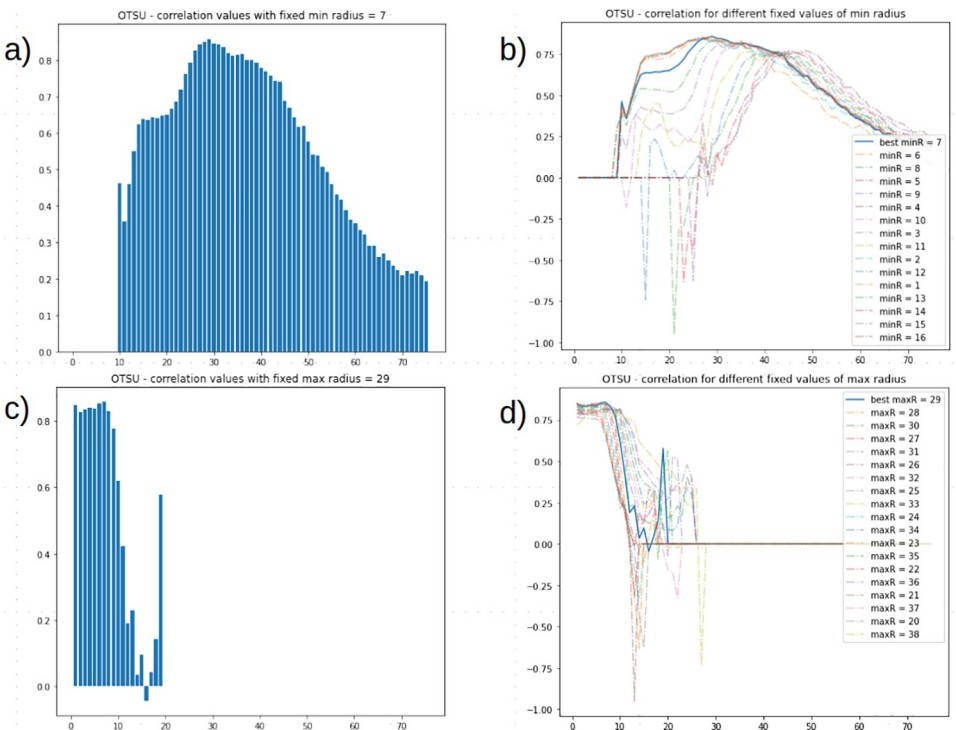

**Fig 4. Correlation values distribution in OTSU's thresholding.** (A-B) Maximum radius, (C-D) minimum radius are singularly ranged.

extracting 50 images and a test set, I_{Test}, containing the remaining 50 images. On these cropped images (1490x678), the normalized image (Fig 6B), a 50% resized image (745x339) and the binary image (Fig 6C) were produced using the latter for detection (Fig 6D). First, pipeline performance was tested, on this dataset, with the best parameters combinations previously obtained. To test the pipeline output in the normalization step, the pipeline was run with Otsu's automatic thresholding method and the Circle Hough Transform with the set of parameters reported in Table 1. The obtained correlations are reported in Table 4. Furthermore, the pipeline output in the thresholding step was tested with the same grid search approach previously used.

In detail, for OTSU thresholding, the grid search approach was performed on I_{Train} with *minRadius* and *maxRadius* ranging from 1 to 75 with a step of 1 (Fig 7). The highest correlation resulted from: *minRadius*—12, *maxRadius*—43 with a correlation of 0.998. Correlation values distribution for best parameters of *minRadius* and *maxRadius* are shown in Fig 7. Then, the pipeline was tested on I_{Test} with *minRadius* set to 12 and *maxRadius* set to 43 yielding a correlation of 0.998.

For the Adaptive thresholding methods, the first grid search approach was performed on I_{Train} with *minRadius* set to 12 (best value found in OTSU), *maxRadius* ranging from 15 to 45 (range was chosen since *maxRadius* was shown, in OTSU in Fig 7, to reach a plateau in range [20,50], with a step of 1, *blocksize* ranging from 3 to 50 with a step of 4, C ranging from 1 to 60 with a step of 4. The highest correlation resulted from: for the AMT method, *maxRadius*—17, C constant—16, *blocksize*—11 with a correlation of 0.999 on I_{Train} and a correlation of 0.996 on I_{Test}; for the AGT method, *maxRadius*—17, C constant—8, *blocksize*—11 with a correlation of 0.999 on I_{Train} and a correlation of 0.999 on I_{Test}.

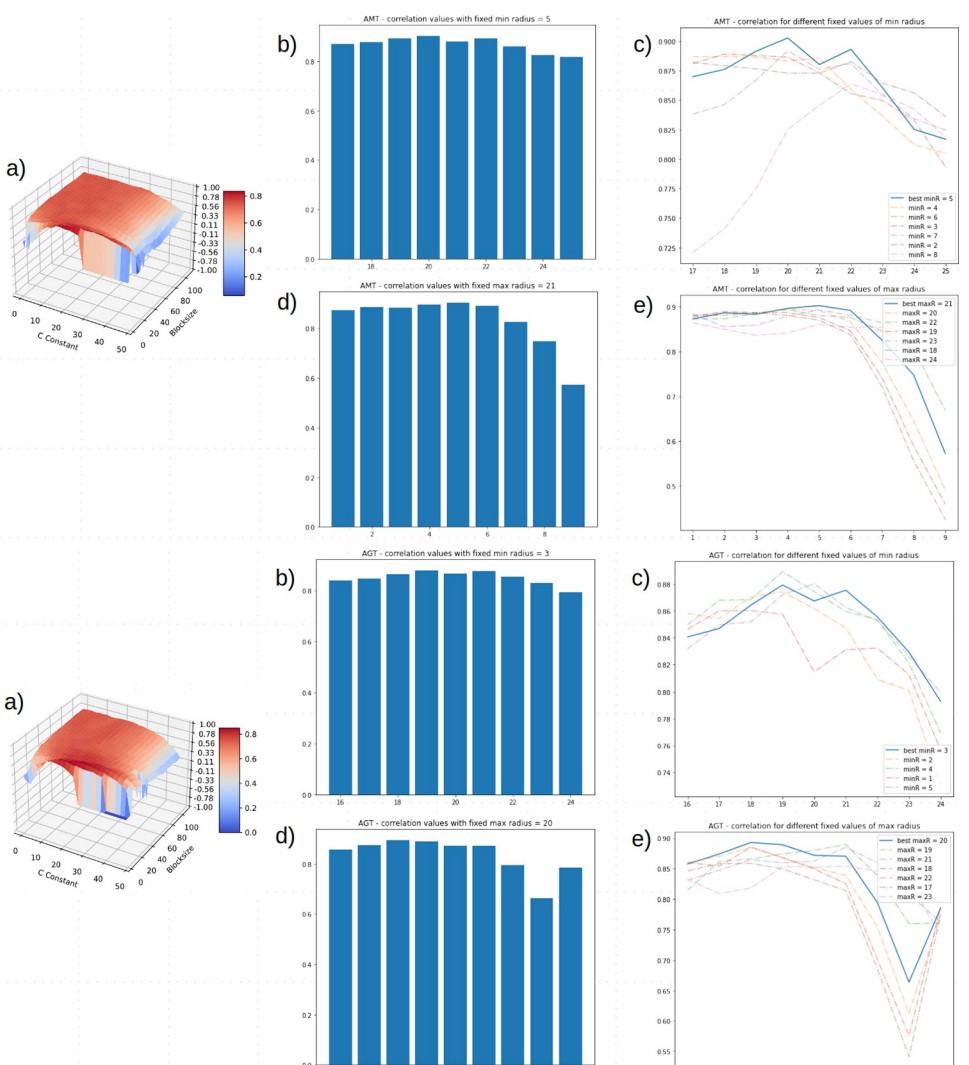

**Fig 5. Correlation values distribution in AMT (top) and AGT (bottom) thresholding when ranging *blocksize*, C Constant, minRadius and maxRadius.** (A) For surface plots, minRadius was initially set to 1 and maxRadius was set to 15 to show the range of best parameters of both *blocksize* and C constant; then, after grid search, both *blocksize* and C were set to the best values (AMT—C = 33—bsize = 9, AGT—C = 27—bsize = 17) to show (B-C) maxRadius and (D-E) minRadius distributions.

The second grid search approach was performed with AMT on I_{Train} with *blocksize* set to 11 and *C* set to 16 (best values found in previous step) and with AGT on I_{Train} with

**Table 3. Pipeline thresholding and CHT best parameters combinations.**

| Thresholding | minRadius | maxRadius | C constant | blocksize | Correlation % |
|---|---|---|---|---|---|
| OTSU (training set) | 7 | 29 | | | 0.856 |
| OTSU (test set) | 7 | 29 | | | 0.819 |
| Adaptive Mean (training set) | 5 | 21 | 33 | 9 | 0.902 |
| Adaptive Mean (test set) | 5 | 21 | 33 | 9 | 0.834 |
| Adaptive Gaussian (training set) | 3 | 20 | 27 | 17 | 0.893 |
| Adaptive Gaussian (test set) | 3 | 20 | 27 | 17 | 0.787 |

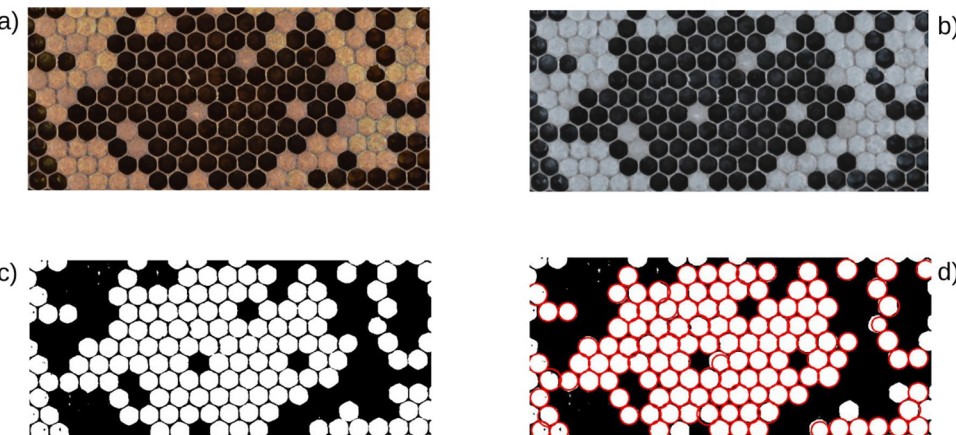

**Fig 6. Image analysis pipeline on DeepBee images.** (A) Original image, (B) normalized image, (C) thresholded image, (D) circle-detected image.

*blocksize* set to 11 and *C* set to 8, while *minRadius* and *maxRadius* both ranged from 1 to 50 with a step of 1. The highest correlation resulted from: for the AMT method, *minRadius*—12, *maxRadius*—17, *C* constant—16, *blocksize*—11 with a correlation of 0.999 on $I_{Train}$ and a correlation of 0.996 on $I_{Test}$; for the AGT method, *maxRadius*—17, *C* constant—8, *blocksize*—11 with a correlation of 0.999 on $I_{Train}$ and a correlation of 0.999 on $I_{Test}$.

The global grid search, involving all four parameters, was performed limiting parameters range to a specific subspace. In particular, in AMT, *minRadius* was set to range from 8 to 16 with a step of 1, *maxRadius* was set to range from 13 to 21 with a step of 1, *blocksize* was set to range from 5 to 13 with a step of 2, *C constant* was set to range from 0 to 18 with a step of 2; in AGT, *minRadius* was set to range from 8 to 16 with a step of 1, *maxRadius* was set to range from 13 to 21 with a step of 1, *blocksize* was set to range from 7 to 15 with a step of 2, *C constant* was set to range from 4 to 22 with a step of 2.

The highest correlation resulted from: for the AMT method, *minRadius*—12, *maxRadius*—17, *C* constant—9, *blocksize*—9 with a correlation of 0.999 on $I_{Train}$ and a correlation of 0.998 on $I_{Test}$; for the AGT method, *minRadius*—12, *maxRadius*—17, *C* constant—13, *blocksize*—11 with a correlation of 0.999 on $I_{Train}$ and a correlation of 0.999 on $I_{Test}$. *minRadius* correlation values distribution when fixing *maxRadius* to the best value obtained (AMT—17, AGT—17) is reported in Fig 8 as well as *maxRadius* correlation values distribution when setting *minRadius* to the best value obtained (AMT -12, AGT—12) using in both cases fixed C constant and *blocksize* best values (AMT—9–9, AGT—13–11). The obtained correlations are reported in Table 5.

**Table 4. DeepBee images pipeline performance in normalization step.**

| Step 1 | Step 2 | Correlation % |
|---|---|---|
| Vahadane | 50% Resizing (745x339) | 0.997 |
| Macenko | 50% Resizing (745x339) | 0.630 |
| ACE | 50% Resizing (745x339) | 0.997 |
| No Normalization | 50% Resizing (745x339 | 0.997 |

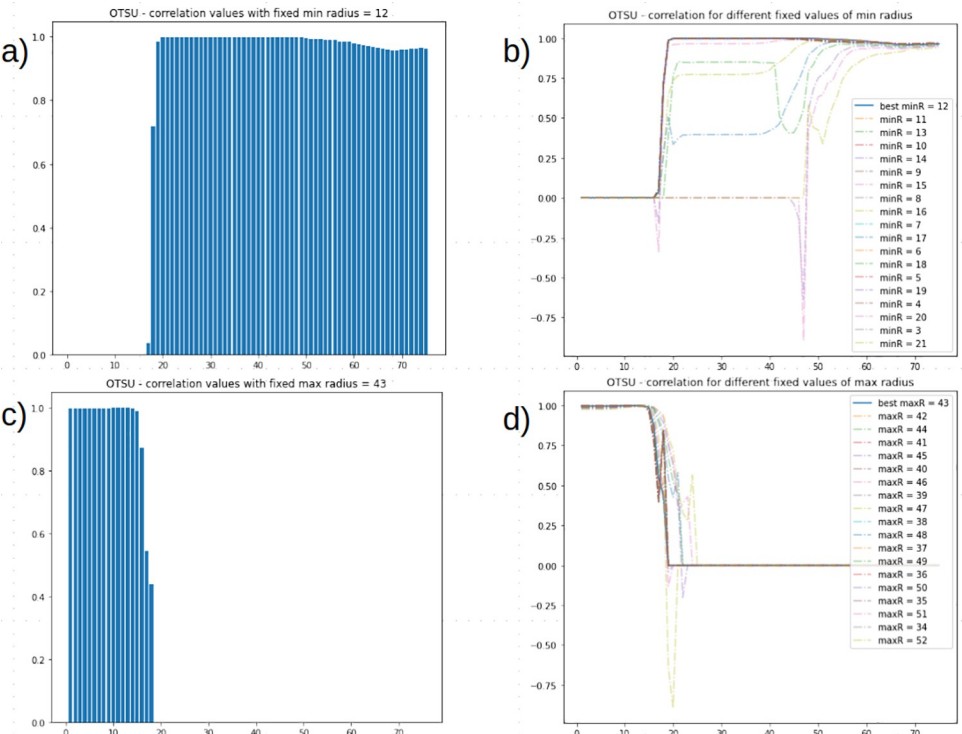

**Fig 7. Correlation values distribution in OTSU's thresholding.** (A-B) Maximum radius, (C-D) minimum radius are singularly ranged.

## Discussion and conclusions

Since hygienic behavior is defined as a response of worker bees of disease spreading in honeybee colony and, when the amount of worker bees showing it is sufficient, a good colony-level resistance is achieved, it is important to analyze and quantify it through the rate of removal of dead broods of a colony.

Usually, hygienic behavior is determined through the pin-killed brood assay [3] or through the freeze-killed brood test [1, 5]. The study proposed in Alves et al. is one of the most recent and promising works proposing a fully automated approach for the detection of capped brood in comb frames in the hive and the classification of seven different comb cell classes [18]. They assured image capture standardization through development of a wooden tunnel sealed for external light and with optimized dimensions. Their approach involved a preprocessing step through the application of a Contrast Limited Adaptive Histogram Equalization (CLAHE) by [26] and a bilateral filter for noise reduction; the detection step involved Circle Hough Transform [25] leading to a detection rate of 98.7%; they classified comb cells through several convolutional neural networks (CNNs).

Focus of this study was to assess hygienic behavior through analysis of images captured by beekeepers in field conditions after the FKB test; due to the nature of the test, it was not possible to standardize image capture leading to presence of uncontrolled illumination, differing color conditions, rotations, scaling and comb sizes. Pipeline performance was assessed correlating manual counted uncapped cells to automatic detected ones. Each step of the pipeline was, progressively, tested to asses both the best algorithm and parameters for detection: first, in the preprocessing step, a manual crop of the freeze-killed brood ROI was produced followed by a 10% resizing; then, salt and pepper noise as well as Gaussian noise were removed through

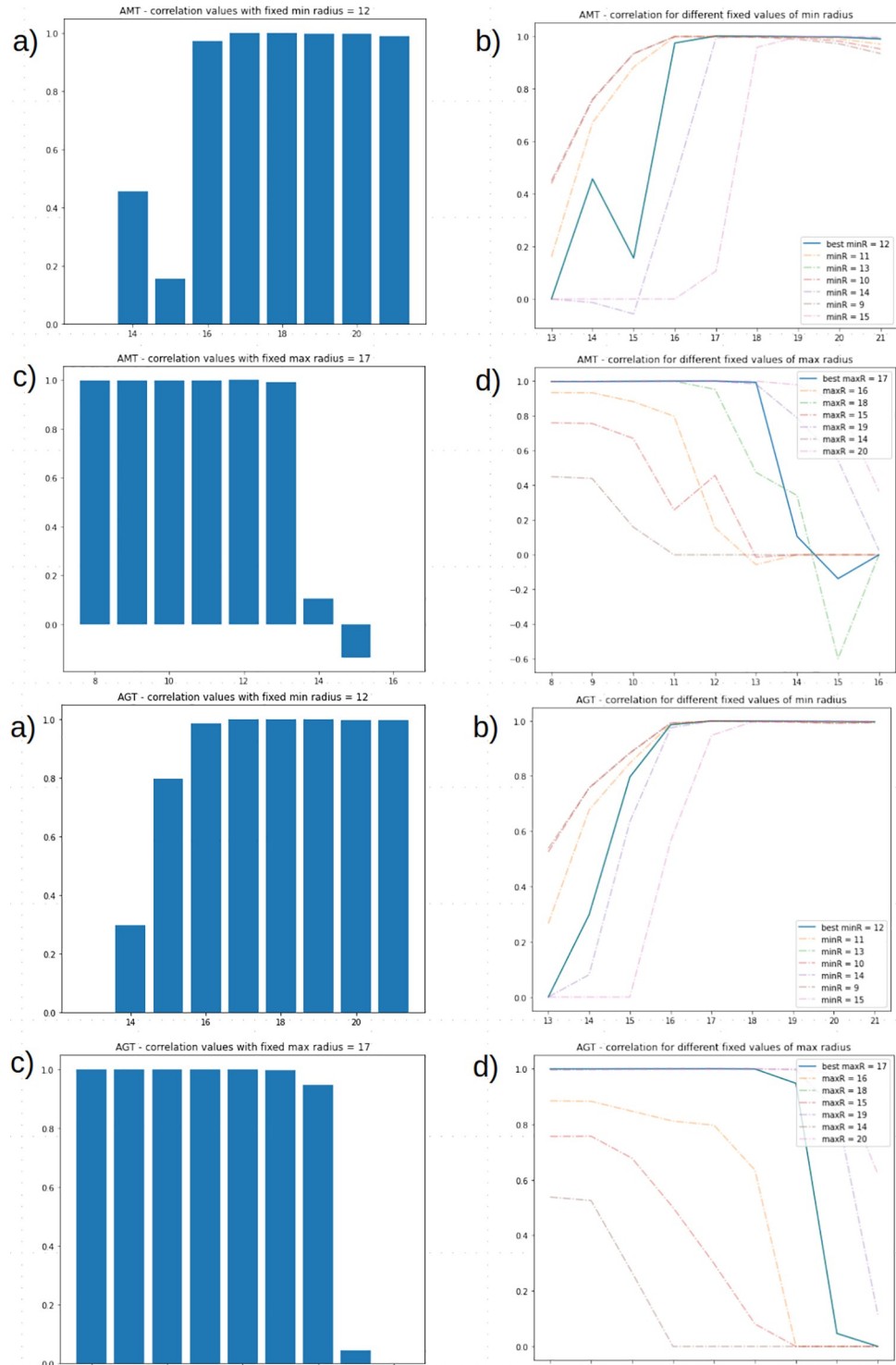

**Fig 8. Correlation values distribution in AMT (top) and AGT (bottom) thresholding when ranging *blocksize*, C Constant, minRadius and maxRadius.** Both *blocksize* and C were set to the best values (AMT—C = 9—bsize = 9, AGT—C = 13—bsize = 11) to show (A-B) maxRadius and (C-D) minRadius distributions.

a median filter followed by a Gaussian filter; last, several color normalization algorithms such

**Table 5. DeepBee images pipeline performance in thresholding step.**

| Thresholding | minRadius | maxRadius | C constant | blocksize | Correlation % |
|---|---|---|---|---|---|
| OTSU (training set) | 12 | 43 | | | 0.998 |
| OTSU (test set) | 12 | 43 | | | 0.997 |
| Adaptive Mean (training set) | 12 | 17 | 9 | 9 | 0.999 |
| Adaptive Mean (test set) | 12 | 17 | 9 | 9 | 0.998 |
| Adaptive Gaussian (training set) | 12 | 17 | 13 | 11 | 0.999 |
| Adaptive Gaussian (test set) | 12 | 17 | 13 | 11 | 0.998 |

as Automatic Color Equalization [19–21], an algorithm developed by [22] in digital histology and a more recent one called Structure-Preserving Color Normalization (SPCN) developed by [23] were explored. Second, in the thresholding step, several algorithms were tested such as OTSU's automatic thresholding [24], Mean Adaptive Thresholding, Gaussian Adaptive Thresholding. Finally, we detected uncapped comb cells through Circle Hough Transform.

To assess the best normalization approach, the pipeline was tested, on the whole dataset, with OTSU's automatic thresholding, since it does not require further parameters tuning, and Circle Hough Transform parameters *minRadius*—1, *maxRadius*—25. The ACE algorithm was found to work best in simultaneously handling poor illuminations and differing color conditions yielding a correlation of 0.825. The SPCN was slightly less performant with a correlation of 0.797 while the Macenko method showed comparable results with a test done with no normalization. When images were resized before normalization, the correlation dropped in the range of 0.439–0.517.

Then, the following steps were tested on 127 images, which were used to compose a training set, *I_{Train}*, by randomly extracting 50 images and a test set, *I_{Test}*, containing the remaining 77 images.

To assess the best thresholding approach, the pipeline was tested with ACE normalization and a combination of Circle Hough Transform parameters: OTSU's thresholding was tested by ranging Circle Hough Transform parameters *minRadius* and *maxRadius* from 1 to 75 with a step of 1. The best parameters combination resulted in *minRadius*—7, *maxRadius*—29 with a correlation of 0.856 on *I_{Train}* and with a correlation of 0.819 on *I_{Test}*. Setting minimum radius to 7 while ranging *maxRadius* showed a normal distribution in correlation values (Fig 4) with a plateau around 29. Setting *maxRadius* to 29 (highest correlation in previous step) while ranging *minRadius* showed similar correlation (*minRadius* 7 had highest correlation) values until correlation dropped considerably when *minRadius* reached 10 (Fig 4). Since the Adaptive thresholding methods (Mean and Gaussian Adaptive Thresholding) introduced tuning of two parameters to determine the threshold value (*blocksize* and *C* constant), a grid search approach involving three parameters (CHT *maxRadius*, *blocksize* and *C* constant) was set up excluding *minRadius* for a grid search with four parameters has a high computational cost. A first grid search for the Adaptive Mean Thresholding and the Adaptive Gaussian Thresholding was performed with a coarse parameter range to identify the 'optimal subspace', a range in which *maxRadius*, *blocksize* and *C* yielded high correlation and with *minRadius*—7, which is best value found in OTSU. Then, a second grid search was performed with fixed *blocksize* and *C* while ranging only CHT radiuses. Finally, a global grid search involving all four parameters was performed limiting their range in the optimal subspace found in previous grid searches. The highest correlation resulted from: for the AMT method, *minRadius*—5, *maxRadius*—21, *C* constant—33, *blocksize*—9 with a correlation of 0.902 on *I_{Train}* and a correlation of 0.834 on *I_{Test}*; for the AGT method, *minRadius*—3, *maxRadius*—20, *C* constant—27, *blocksize*—17 with a correlation of 0.893 on *I_{Train}* and a correlation of 0.787 on

*I_{Test}*. AMT and AGT showed overfitting when run on *I_{Train}* considering their drop in performance when run on *I_{Test}*.

To assess the performance of the developed pipeline on an independent image set, we sampled 100 images from Alves et al. [18] in which a rectangular area was cropped. Manual uncapped cell counts were generated from each sampled image which were used as reference for pipeline-generated counts. For the normalization step, the pipeline was run with OTSU's thresholding and Circle Hough Transform minimum radius—1 and maximum radius—25; both ACE and the Vahadane method had a correlation of 0.997. The Macenko method was shown to have the lowest correlation 0.630. It is worth noting that, with this dataset, performing a detection with no previous normalization resulted in a correlation of 0.997. For the thresholding step, the pipeline was run, first, with ACE normalization and OTSU's thresholding tested by ranging Circle Hough Transform parameters *minRadius* and *maxRadius* from 1 to 75 with a step of 1, resulting in the best parameters *minRadius*—12, *maxRadius*—43 with a correlation of 0.998 on *I_{Train}* and *I_{Test}*. Both the Mean Adaptive Thresholding and the Gaussian Adaptive Thresholding, run with the best parameters obtained in the grid search approach in the testing phase and reported in Table 5, yielded slightly superior results. To set an even better setting in terms of lighting condition and detection, as well as a comparable setting to the images from the public dataset [18], the developed pipeline was run on 100 images sampled from our pool after cropping a rectangular area from the ROI of the FKB brood test (S1 Fig). The obtained correlations are reported in S1 and S2 Tables. The increase in detection rates was attributed to differing image capture settings. In all of the performed tests, normalization with ACE coupled with Otsu's thresholding yields comparable results when coupled with AMT and AGT while not requiring further parameter tuning. In conclusion, our results show that the image processing strategy we are proposing successfully handles a broad range of image illuminations and exposures, and it may be therefore used to avoid impractical, time-consuming, and sometimes even costly image acquisition setups. We tested our model on the count of uncapped cells from honeybee comb images, as requested by beekeepers assessing hygienic behavior through the FKB. The comparative evaluation of our pipeline on the private dataset acquired in the field by beekeepers and on a dataset composed of images from the public dataset provided by Alves et al. [18] shows that the results may be further improved if the image exposure is controlled.

Of note, the presented pipeline is aimed at identifying and counting the uncapped comb cells. Another important problem is the detection of larvae or eggs in uncapped comb cells.

Therefore, future work will be aimed at extending our pipeline to differentiate empty uncapped cells, uncapped cells containing larvae, and uncapped cells containing eggs.

## Supporting information

**S1 Fig. Image analysis pipeline on 100 cropped images.** Original crop image a), normalized image b), thresholded image c), circle-detected image d).
(TIF)

**S1 Table. 100 cropped images pipeline performance in normalization step.**
(XLSX)

**S2 Table. 100 cropped images pipeline performance in thresholding step.**
(XLSX)

## Acknowledgments

We thank bee breeders for providing images.

## Author Contributions

**Conceptualization:** Stefano Biffani.

**Data curation:** Gianluigi Paolillo, Giulietta Minozzi.

**Formal analysis:** Gianluigi Paolillo, Elena Casiraghi.

**Funding acquisition:** Giulietta Minozzi.

**Investigation:** Maria Grazia De Iorio.

**Methodology:** Alessandro Petrini.

**Software:** Gianluigi Paolillo, Alessandro Petrini, Elena Casiraghi.

**Supervision:** Elena Casiraghi, Giorgio Valentini.

**Validation:** Gianluigi Paolillo, Elena Casiraghi.

**Writing – original draft:** Gianluigi Paolillo, Elena Casiraghi, Maria Grazia De Iorio, Giulietta Minozzi.

**Writing – review & editing:** Gianluigi Paolillo, Elena Casiraghi, Stefano Biffani, Giulio Pagnacco, Giulietta Minozzi, Giorgio Valentini.

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
