## [Decision Letter · Decision Letter 0]

3 Jan 2022

PONE-D-21-35191Automated image analysis of Varroa related traits in honeybee comb imagesPLOS ONE

Dear Dr. Casiraghi,

Thank you for submitting your manuscript to PLOS ONE. After careful consideration, we feel that it has merit but does not fully meet PLOS ONE’s publication criteria as it currently stands. Therefore, we invite you to submit a revised version of the manuscript that addresses the points raised during the review process (for details, please see below).

Please submit your revised manuscript by Feb 17 2022 11:59PM If you will need more time than this to complete your revisions, please reply to this message or contact the journal office at plosone@plos.org. Please include the following items when submitting your revised manuscript:A rebuttal letter that responds to each point raised by the academic editor and reviewer(s). You should upload this letter as a separate file labeled 'Response to Reviewers'.A marked-up copy of your manuscript that highlights changes made to the original version. You should upload this as a separate file labeled 'Revised Manuscript with Track Changes'.An unmarked version of your revised paper without tracked changes. You should upload this as a separate file labeled 'Manuscript'.

We look forward to receiving your revised manuscript.

Kind regards,

Wolfgang Blenau

Academic Editor

PLOS ONE

Reviewers' comments:

Reviewer's Responses to Questions

**Comments to the Author**

1. Is the manuscript technically sound, and do the data support the conclusions?

Reviewer #1: Yes

2. Has the statistical analysis been performed appropriately and rigorously? 

Reviewer #1: Yes

3. Have the authors made all data underlying the findings in their manuscript fully available?

Reviewer #1: Yes

4. Is the manuscript presented in an intelligible fashion and written in standard English?

Reviewer #1: Yes

5. Review Comments to the Author

Reviewer #1: In the manuscript “Automated image analysis of Varroa related traits in honeybee comb images” a new approach for the automated counting of open/capped brood cells of honeybees is presented and applied to images from a freeze killed brood (FKB). The manuscript presents a new method and is definitively worth publishing, however, the title and abstract seem to fit very well to the content of the manuscript, as title and abstract suggest that the paper focusses on analyses of FKB test in relation to Varroa infestation, while in fact the paper is a very technical manuscript focussing almost exclusively on image recognition algorithms. Therefore, I’d suggest to change the title and rewrite the abstract to make the clearer what the paper is about. I would also mention they highlight of the new method in the abstract, i.e. the use of methods which can cope with very varying image quality or exposure.

Specific comments:

Line 29: The sentence “Focus of this study is to design an automated pipeline for segmentation and automatic count of honeybee comb images acquired during summer 2020 by beekeepers.” is not clear to me. What do you mean with segmentation? And I guess you don’t want to count images, but maybe cells?

Line 34: What is meant with “To recover from acquisition”? Do you mean for image optimisation? Or for normalisation?

Line 37: From the beginning of the abstract I expected that varroa infestation is addressed. From the sentence here it seems that only cells are counted. That is confusing. What is the subject of the paper?

Lines 39 and 43: What is the difference from the 77 ”unseen” images and the 100 randomly extracted images? Both were taken from the set of 127 images after removal of the images used for training (but then the numbers wouldn’t add up)? Or did you mix images used for training and images not used for training in the set of 100 images? If so, that would not be suitable for validation. I suggest to clarify this section.

Line: 47: It seems that indeed only cells are counted. Then I find the title a bit misleading. The title could be e.g. “Automated image analysis to assess hygienic behaviour of honeybees”.

Line 68: The list of references given could be more comprehensive. There are more papers on counting brood and uncapped cells and papers where such methods have been applied: Jeker et al. 2012, Avni et al. 2015, Cutler et al. 2014, Collin et al 2018, Wang er al. 2020, etc.

Line 81: Controlled conditions are generally used in order to optimise results in particular when doing large scale analyses. It is not so much the problem that automated analysis doesn’t work well if conditions are not controlled, but that manual adjustment of images is not needed when using controlled conditions. Hence it is more a practical point to reduce work load. Furthermore, for counting cells all software solutions seem to work pretty well even under varying conditions. Detecting larvae or eggs is the main problem in practice.

Line 82: I disagree that it is difficult to do in practice. Most researchers use a setup with artificial light for photography. With that it is easy to produce many images with identical exposure and size of frames in images. Routine analyses can include hundreds to thousands of standardised images. This has also been done in some of the references given above.

Line 100: Just a comment, the paper addresses a method to make counting less time consuming. In the study presented in the manuscript images were manually cropped. Would existing software have been used, then both cropping and automated counting could have been done within the software much less time effort.

Lines 95-389: A general comment on “Materials and methods” and “Results”. These sections are very, very detailed and extremely difficult to follow. I expect that only a handfull of readers will be able to understand all detail. The broad majority of those working on honeybees probably want to understand the key principles, but that is difficult to understand for a non-expert in image recognition. The best summary of the paper is figure 1. I would suggest to make these key principles clearer in the text and also to indicate clearly when technical details start (so a non-expert can skip them) and when they end. This could be done by an introductory paragraph in each section, which explains only the key principle in common language and going into detail only after that.

Line 105: Now it becomes clear what the 100 image set mentioned in the abstract referred to.

Line 115: Please indicate version numbers of software or libraries.

Line 396: Maybe I misunderstood the sentence, but I don’t think Alves et al (2020) is first study using completely automatic digital photography. It is being used since over 10 years. Publications are available.

Line 405: After reading the methods and result, my impression is that the method has been used to exemplarily assess hygienic behaviour. However, the key finding is the development of a new method based on Hugh transforms and normalisation of images in order to be able to use images with very varying quality or exposure. This is certainly worth publishing, but the title and abstract makes the reader expect a very different paper. Hence I suggest to make clear (in both the title and abstract) that this paper is on a new methodology and that the focus is to use non-standardised images. Then in the methods and results it could be mentioned that it is tested exemplarily on images of a FKB test.

Another question is, why have many rather standardised images been used for validation if this method is proposed to handle images with very variable exposure? This should also be discussed.

6. PLOS authors have the option to publish the peer review history of their article (what does this mean?). If published, this will include your full peer review and any attached files.

Reviewer #1: No

---

## [Author Response · Author response to Decision Letter 0]

6 Jan 2022

Dear Editor and Reviewer,

We thank you for the interest in our work and for your notes that surely helped us to improve the scientific quality of our manuscript.

Please find below detailed answers to your comments.

Yours sincerely

elena casiraghi (on behalf of all authors)

------

Reviewer's Responses to Questions

1. Is the manuscript technically sound, and do the data support the conclusions?

Reviewer #1: Yes

2. Has the statistical analysis been performed appropriately and rigorously?

Reviewer #1: Yes

3. Have the authors made all data underlying the findings in their manuscript fully available?

Reviewer #1: Yes

4. Is the manuscript presented in an intelligible fashion and written in standard English?

Reviewer #1: Yes

Reviewer #1: In the manuscript “Automated image analysis of Varroa related traits in honeybee comb images” a new approach for the automated counting of open/capped brood cells of honeybees is presented and applied to images from a freeze killed brood (FKB). The manuscript presents a new method and is definitively worth publishing, however, the title and abstract seem to fit very well to the content of the manuscript, as title and abstract suggest that the paper focuses on analyses of FKB test in relation to Varroa infestation, while in fact the paper is a very technical manuscript focussing almost exclusively on image recognition algorithms. Therefore, I’d suggest changing the title and rewriting the abstract to make it clearer what the paper is about. I would also mention the highlight of the new method in the abstract, i.e. the use of methods which can cope with very varying image quality or exposure.

Authors: We would like to thank the Reviewer for his notes. We have carefully revised our manuscript to address all his/her concerns. Briefly, as the Reviewer suggested we have revised both the title and abstract to make clear that we are presenting an automated cell counting method whose strength is the ability of coping with varying image quality or exposure.

Please, find below the detailed answers to all the Reviewers’ concerns.

 

Specific comments:

Line 29: The sentence “Focus of this study is to design an automated pipeline for segmentation and automatic count of honeybee comb images acquired during summer 2020 by beekeepers.” is not clear to me. What do you mean with segmentation? And I guess you don’t want to count images, but maybe cells?

Authors: we agree with the Reviewer the the sentence was not clear. In the digital image processing field, “segmentation” refers to the automatic identification of objects in the image. We changed the sentence to better clarify that our aim is the identification of capped brood cells in honeybee comb images, which ultimately allows their automatic counting.

Line 34: What is meant with “To recover from acquisition”? Do you mean for image optimisation? Or for normalisation?

Authors: we tested three different image enhancement algorithms to choose the one that mostly reduced the effect of bad illumination conditions or exposures. We agree with the Reviewer that our sentence was not clear and we therefore substituted that sentence with the following one:

“[...] To reduce the effects of varied illuminations or exposures, three image enhancement algorithms were tested and compared […]”

Line 37 and Line 47: From the beginning of the abstract I expected that varroa infestation is addressed. From the sentence here it seems that only cells are counted. That is confusing. What is the subject of the paper?

It seems that indeed only cells are counted. Then I find the title a bit misleading. The title could be e.g. “Automated image analysis to assess hygienic behaviour of honeybees”.

Authors: The subject of the paper is the automatic count of capped brood cells in honeybee comb images acquired under varied illuminations and exposures and we agree with the Reviewer that the title we used was misleading while the one She/He suggests is more appropriate. We therefore changed the title as suggested by the Reviewer.

Lines 39 and 43: What is the difference from the 77 ”unseen” images and the 100 randomly extracted images? Both were taken from the set of 127 images after removal of the images used for training (but then the numbers wouldn’t add up)? Or did you mix images used for training and images not used for training in the set of 100 images? If so, that would not be suitable for validation. I suggest to clarify this section.

Authors: we are sorry but indeed the abstract was confusing. Our dataset contains 127 images acquired under various illuminations and exposures. To obtain an unbiased evaluation, we randomly split our dataset to obtain a training image set (composed of 50 images) and a test image set (composed of 77 images).

The training images have been used for developing the pipeline and for parameter tuning, that is for choosing the parameters values that allow to obtain the best performance on the training set.

The test image set allows obtaining an unbiased estimate because the images in the test set were never used during the pipeline development or the parameter tuning phase. This is why they are generally called “unseen” in the artificial intelligence field. Anyhow, to avoid confusion, we have removed the adjective “unseen” in the revised abstract.

On the other hand, the dataset composed of 100 images is a completely different dataset that has been randomly extracted from the publicly available dataset by Alves et al. We have used it for assessing our model against a publicly available dataset.

We have rewritten the Abstract to remove all the confusing sentences. More precisely, the following sentences summarize how we split our dataset into training and test sets and that we also tested our model on an external, dataset randomly extracted from a publicly available dataset:

[...] To obtain an unbiased evaluation, the cropped images were randomly split into a training image set (50 images), which was used to develop and tune the proposed model, and a test image set (77 images), which was solely used to test the model. [...]

When applied to the 77 test images the model obtained a correlation of 0.819 between the automated counts and the experts’ counts. We further assessed the proposed model on 100 images randomly extracted from a public dataset acquired under controlled conditions. On this new set, the correlation with manually counted cells on the dataset was much higher (0.997) than the one we obtained on our dataset. [...]

Line 68: The list of references given could be more comprehensive. There are more papers on counting brood and uncapped cells and papers where such methods have been applied: Jeker et al. 2012, Avni et al. 2015, Cutler et al. 2014, Colin et al 2018, Wang er al. 2020, etc.

Authors: we thank the reviewer and we have added all the above mentioned references in the introduction of the revised manuscript.

Line 81: Controlled conditions are generally used in order to optimise results in particular when doing large scale analyses. It is not so much the problem that automated analysis doesn’t work well if conditions are not controlled, but that manual adjustment of images is not needed when using controlled conditions. Hence it is more a practical point to reduce workload. 

Line 82: I disagree that it is difficult to do in practice. Most researchers use a setup with artificial light for photography. With that it is easy to produce many images with identical exposure and size of frames in images. Routine analyses can include hundreds to thousands of standardised images. This has also been done in some of the references given above.

Authors: we agree with the Reviewer.However, the poor results we obtained on our images when running preliminary experiments with CombCount method (Colin at al., 20218 [17]) and the method from Alves et al., 2020 [18] showed that the usage of predefined acquisition settings results in ad-hoc techniques specifically developed for handling the illumination and exposure conditions defined by the acquisition settings. 

We have substituted the sentence at lines 81-82 to clear this point.

Line 81: Furthermore, for counting cells all software solutions seem to work pretty well even under varying conditions. Detecting larvae or eggs is the main problem in practice.

Authors: We agree with the Reviewer and indeed this is one of our future works. To clear this point we have inserted a sentence at the end of the discussion section.

Line 100: Just a comment, the paper addresses a method to make counting less time consuming. In the study presented in the manuscript images were manually cropped. Would existing software have been used, then both cropping and automated counting could have been done within the software much less time effort.

Authors: as mentioned above, before developing the model we are presenting we performed preliminary experiments with CombCount [17] and the method from Alves et al. [18]. Unfortunately, both the methods provided poor cropping and automated counting results. This is probably due to the fact that our images present different illuminations and exposures. 

Lines 95-389: A general comment on “Materials and methods” and “Results”. These sections are very, very detailed and extremely difficult to follow. I expect that only a handfull of readers will be able to understand all detail. The broad majority of those working on honeybees probably want to understand the key principles, but that is difficult to understand for a non-expert in image recognition. The best summary of the paper is figure 1. I would suggest to make these key principles clearer in the text and also to indicate clearly when technical details start (so a non-expert can skip them) and when they end. This could be done by an introductory paragraph in each section, which explains only the key principle in common language and going into detail only after that.

Authors: we must really thank the Reviewer for the note because it allowed us to greatly improve readability.

To this aim, at the beginning of the “Methods” subsection and of the “Result” section we have inserted introductory paragraphs using common language to summarize the key principles and the technical details that can be found in the section. In this way, non-expert readers may decide whether to skip the Section or to read through it. 

Line 115: Please indicate version numbers of software or libraries.

Authors: we used version 3.7 of Python and the OpenCv3 v.4.0 package. We indicated those versions in the manuscript. 

Line 396: Maybe I misunderstood the sentence, but I don’t think Alves et al (2020) is the first study using completely automatic digital photography. It is being used since over 10 years. Publications are available.

Authors: the sentence may have been misleading. We rephrased it to make it clear that “[...] The study proposed in Alves et al. is the most recent and promising work proposing a fully automated approach for the detection of capped brood in comb frames in the hive and the classification of seven different comb cell classes [14] [...]”

Line 405: After reading the methods and result, my impression is that the method has been used to exemplarily assess hygienic behaviour. However, the key finding is the development of a new method based on Hugh transforms and normalisation of images in order to be able to use images with very varying quality or exposure. This is certainly worth publishing, but the title and abstract makes the reader expect a very different paper. Hence I suggest to make clear (in both the title and abstract) that this paper is on a new methodology and that the focus is to use non-standardised images. Then in the methods and results it could be mentioned that it is tested exemplarily on images of a FKB test.

Another question is, why have many rather standardised images been used for validation if this method is proposed to handle images with very variable exposure? This should also be discussed.

Authors: 

We thank the Reviewer for its important note. To clear this point we changed the title as suggested in a previous note, we completely revised the Abstract and we also modified the end of the Discussion section to highlight the strengths of our method, by pointing out that the count of uncapped cells from comb images is an application that shows the ability of our strategy to handle uncontrolled image conditions. 

Regarding the experiment on the 100 images randomly extracted from the dataset made publicly available from Alves et al. [14], we provide this further test for two reasons: 

1) we wanted to assess the generalizability of our model with respect to the usage of different acquisition equipment and conditions; to this aim, we chose the dataset from Alves et al. [14] because it is the most recent, publicly available dataset, and contains images with varied illuminations even though the image acquisition and exposure are controlled; 

2) we believe it is fair to provide results on a set of images that can be also downloaded by the readers. 

We cleared this point in the Abstract, in the Methods section and in the Discussion Section.

Again, we would like to thank the Reviewer and Editor for the interest in our work.

---

## [Decision Letter · Decision Letter 1]

12 Jan 2022

PONE-D-21-35191R1Automated image analysis to assess hygienic behaviour of honeybeesPLOS ONE

Dear Dr. Casiraghi,

Thank you for submitting your manuscript to PLOS ONE. After careful consideration, we feel that it has merit but does not fully meet PLOS ONE’s publication criteria as it currently stands. Therefore, we invite you to submit a revised version of the manuscript that addresses the points raised during the review process. If you could please react briefly to the remaining comment from Reviewer #1 (see below), I can most likely accept the manuscript without involving reviewers again.

We look forward to receiving your revised manuscript.

Kind regards,

Wolfgang Blenau

Academic Editor

PLOS ONE

Journal Requirements:

Reviewers' comments:

Reviewer's Responses to Questions

**Comments to the Author**

1. If the authors have adequately addressed your comments raised in a previous round of review and you feel that this manuscript is now acceptable for publication, you may indicate that here to bypass the “Comments to the Author” section, enter your conflict of interest statement in the “Confidential to Editor” section, and submit your "Accept" recommendation.

Reviewer #1: All comments have been addressed

2. Is the manuscript technically sound, and do the data support the conclusions?

Reviewer #1: Yes

3. Has the statistical analysis been performed appropriately and rigorously? 

Reviewer #1: Yes

4. Have the authors made all data underlying the findings in their manuscript fully available?

Reviewer #1: Yes

5. Is the manuscript presented in an intelligible fashion and written in standard English?

Reviewer #1: Yes

6. Review Comments to the Author

Reviewer #1: I’d like to thank the authors for the work they put into the revision. The readability and clarity has significantly been improved, which now makes the manuscript more accessible to a wider audience.

There is only one minor comment:

On Page 4 you compare several existing methods for the automatic counting of brood cells. However, in your response to the reviewers comments you mentioned that you only tested CombCount (Colin at al., 20218 [17]) and the method from Alves et al. (2020 [18]). Hence it is not clear how you concluded on the difficulties of the other methods, if you haven’t used these. I would suggest to rephrase the sentence making that clear, e.g. by writing the pipeline by Jeker “seems to” be laborious or IndiCounter “seems to” to depend on specific acquisition conditions (our observation is, by the way, different: We find that in general automated recognition works better when image is ok and in particular capped brood cells are recognised well with all kinds of software, without “specific acquisition conditions”; capped cells are well recognised even in very low-res images from low-end cameras of phones). You could rephrase the sentence to:

“Many software packages, able to evaluate comb frames are available. Some of them [11,12] perform statistical analysis to study the condition of honeybee colony by using commercial softwares (“IndiCounter, WSC Regexperts” available at https://wsc-regexperts.com/en/software-and-databases/software/honeybee-brood-colonyassessment-software/) which seems to be designed for larger scale studies where specific acquisition conditions or often used. On the other hand, the semi-automated pipeline introduced by Jeker et al. [13] seems to requires a laborious acquisition setting that depends on several camera parameters to be carefully set.”

7. PLOS authors have the option to publish the peer review history of their article (what does this mean?). If published, this will include your full peer review and any attached files.

Reviewer #1: No

---

## [Author Response · Author response to Decision Letter 1]

13 Jan 2022

Dear Editor and Reviewer,

We thank you for the interest in our work and for your notes that surely helped us to improve the scientific quality of our manuscript.

Please find below detailed answers to your comments.

Yours sincerely

elena casiraghi (on behalf of all authors)

Review Comments to the Author

Reviewer #1: I’d like to thank the authors for the work they put into the revision. The readability and clarity has significantly been improved, which now makes the manuscript more accessible to a wider audience.

There is only one minor comment:

On Page 4 you compare several existing methods for the automatic counting of brood cells. However, in your response to the reviewers comments you mentioned that you only tested CombCount (Colin at al., 20218 [17]) and the method from Alves et al. (2020 [18]). Hence it is not clear how you concluded on the difficulties of the other methods, if you haven’t used these. I would suggest to rephrase the sentence making that clear, e.g. by writing the pipeline by Jeker “seems to” be laborious or IndiCounter “seems to” to depend on specific acquisition conditions (our observation is, by the way, different: We find that in general automated recognition works better when image is ok and in particular capped brood cells are recognised well with all kinds of software, without “specific acquisition conditions”; capped cells are well recognised even in very low-res images from low-end cameras of phones). You could rephrase the sentence to:

“Many software packages, able to evaluate comb frames are available. Some of them [11,12] perform statistical analysis to study the condition of honeybee colony by using commercial softwares (“IndiCounter, WSC Regexperts” available at https://wsc-regexperts.com/en/software-and-databases/software/honeybee-brood-colonyassessment-software/) which seems to be designed for larger scale studies where specific acquisition conditions or often used. On the other hand, the semi-automated pipeline introduced by Jeker et al. [13] seems to requires a laborious acquisition setting that depends on several camera parameters to be carefully set.”

Author’s answer: we agree with the Reviewer’s note and we thank him for suggesting a new sentence, which we used to substitute our old sentence.

Thanks again for the interest in our work.

Yours sincerely

elena casiraghi on behalf of all the authors

---

## [Editor Report · Decision Letter 2]

14 Jan 2022

Automated image analysis to assess hygienic behaviour of honeybees

PONE-D-21-35191R2

Dear Dr. Casiraghi,

We’re pleased to inform you that your manuscript has been judged scientifically suitable for publication and will be formally accepted for publication once it meets all outstanding technical requirements.

Kind regards,

Wolfgang Blenau

Academic Editor

PLOS ONE
---

## [Editor Report · Acceptance letter]

19 Jan 2022

PONE-D-21-35191R2 

Automated image analysis to assess hygienic behaviour of honeybees 

Dear Dr. Casiraghi:

I'm pleased to inform you that your manuscript has been deemed suitable for publication in PLOS ONE. Congratulations! Your manuscript is now with our production department. 

Kind regards, 

on behalf of

Dr. Wolfgang Blenau 

Academic Editor

PLOS ONE